# Child food poverty prevalence and its associated factors in Dale district, Sidama region, Ethiopia

Zenamlak Yemane[1], Assefa Philipos Kare[1,2], Amelo Bolka [1,2], Fentaw Wassie Feleke [2,3]*

1 School of Public Health, Yirgalem Hospital Medical College, Yirgalem, Ethiopia, 2 Departemt of Human Nutrition, School of Human Nutrition and Food Science Technology, Hawassa University, Hawassa, Ethiopia, 3 Department of Public Health, School of Public Health, Woldia University, Woldia, Ethiopia

* fentawwassie@gmail.com

## Abstract

### Background

More than one-third of all deaths in children under five are attributable to malnutrition brought on by child food poverty (CFP) practices, while proper feeding techniques are essential for enhancing nutritional status and guaranteeing child survival. There is, however, little research on inadequate feeding practices, especially in the Sidama Region.

### Objective

To assess the magnitude of child food poverty and its associated factors among children aged 6–23 months in Dale district, Sidama region, Ethiopia, 2024.

### Materials and methods

A community-based cross-sectional study was conducted. A multi-stage random sampling technique was applied to select 509 study participants. Data were collected using structured questionnaires through face-to-face interviews. Data quality was assured by pre-testing, training, and questionnaire coding. Bivariable and multivariable logistic regression was done to identify the determinants. Variables having a value P≤ 0.25 during bivariable analysis were entered into multiple logistic regression models. In multivariable analysis, variables with p<0.05 were considered significantly associated.

### Results

CFP prevalence was 30% (95% CI: 26-34). The minimum meal frequency was 77%. Minimum acceptable diet was 59%. Factors significantly associated with CFP prevalence were child age 6 - 11 months ((Adjusted Odds Ratio(AOR)=3.022, 95% CI: 1.813-5.037)), female children (AOR=1.783, 95% CI: 1.123-2.830), number of

**Data availability statement:** All relevant data are within the manuscript and its Supporting Information files.

**Funding:** The author(s) received no specific funding for this work.

**Competing interests:** The authors have declared that no competing interests exist.

**Abbreviations:** ANC, Antenatal Care; AOR, Adjusted Odd Ratio; CF, Complementary Feeding; CFPP, Child Food Poverty Prevalence; CI, Confidence Interval; DHS, Demographic Health Survey; EBF, Exclusive Breastfeeding; EDHS, Ethiopian Demographic and Health Survey; IDA, Inadequate Acceptable Diet; IDD, Inadequate Dietary Diversity; IMF, Inadequate Meal Frequency; IYCF, Infant and Young Child Feeding; ISSS, Introduction of Solid, Semi-Solid or Soft Foods; MAD, Minimum Acceptable Diet; MDD, Minimum Dietary Diversity; MDGs, Millennium Development Goals; MMF, Minimum Meal Frequency; PNC, Postnatal Care; SPSS, Statistical Package for the Social Sciences; UNICEF, United Nations International Children's Emergency Fund; WHO, World Health Organization.

antenatal care follow-up below four times (AOR=3.52, 95% CI: (1.429-4.152), family size more than five in the household(AOR= 3.715, 95% CI: 1.974-6.993), and maternal inadequate knowledge of CFP (AOR= 5.148, 95% CI: 3.146-8.426).

## Conclusion

The prevalence of CFP, which remains a significant public health concern, was 30% in the study area. Child age, child sex, antenatal care follow-ups, inadequate maternal knowledge to feed children, and household size greater than five were significantly associated with child food poverty. Enhancing antenatal care services and raising awareness of the best feeding practices, especially for younger infants and female children should be the main goals of nutrition interventions.

## Introduction

CFP is defined as the inability of children to access and consume a nutritious, diverse, and adequate diet during their first five years of life. Children who eat three to four food categories a day are considered to be in moderate child food poverty, but children who eat just two food groups are considered to be in severe child food poverty [1]. The main causes of this severe CFP include inadequate feeding habits, food instability, household poverty, and restricted access to nutrient-dense foods [2]. Inadequate food consumption at this age resulted in the risk of infections, undernutrition, and developmental delays, which feeds the cycle of poverty and illness that lasts for generations [3].

According to the United Nations International Children's Emergency Fund (UNICEF), 440 million children under five worldwide experience food poverty, with 181 million facing severe deprivation. South Asia bears a disproportionate burden of child food poverty (130 million), with West and Central Africa (73 million) being among the most affected regions. In these areas, high rates of poverty, limited access to nutritious foods, and environmental challenges such as droughts and climate change worsen food insecurity. Like many less-developed nations, Ethiopia has a high rate of undernutrition, with 46% of its children under five living in extreme CFP [1].

The basis for immunological function, cognitive development, and physical growth is laid by nutrition throughout the first two years of life [4]. Stunting, wasting, underweight, and micronutrient deficiencies can result from inadequate dietary diversity and poor feeding practices during this time, which can have long-term effects on economic productivity, education, and health [3]. The World Health Organization (WHO) recommends that children aged 6–23 months consume a minimum acceptable diet (MAD) both minimum dietary diversity (MDD) and minimum meal frequency (MMF) [5]. MMF emphasizes how frequently infants of various ages are offered meals other than breast milk and breastfeeding and is used as a stand-in for assessing a child's energy needs [6,7]. Depending on their individual preferences, non-breastfed children should be fed four or five meals a day, consisting of solid or semisolid feeds in

addition to milk feeds, and one or two snacks [8–12]. MDD referece to regardless of portion size and meal frequency, it indicates that a child has consumed at least five different food groups from the seven WHO-recommended food groups 24 hours before to the survey [11,12]. However, many children in resource-limited settings, including Ethiopia, fail to meet these recommendations due to systemic barriers [13].

In Ethiopia, child food poverty is driven by a complex interplay of factors, including household poverty, food insecurity, and limited access to healthcare services [14]. Economic disparities and high poverty restrict household purchasing power, making it difficult for families to afford nutritious foods [15]. Furthermore, environmental factors such as climate change, droughts, and natural disasters disrupt food production and supply chains, further worsening food insecurity [16,17]. These challenges are particularly pronounced in agrarian communities like those in the Sidama Region, where livelihoods are heavily dependent on subsistence farming [18].

Maternal and caregiver practices play a pivotal role in determining child nutrition outcomes in the country [19]. Child food poverty feeding practices, such as delayed introduction of complementary foods, inadequate meal frequency, and reliance on monotonous diets, contribute significantly to CFP [20]. In many cases, mothers and or caregivers having children under two years of children have inadequate knowledge and optimal feeding practices or face cultural barriers that limit their ability to provide diverse and nutritious diets [21]. Improving child nutrition requires filling in these knowledge gaps and encouraging behavioral change [22] on the modified eight food groups for children under two year [11,12].

The proliferation of unhealthy food environments, characterized by the widespread availability of ultra-processed foods and aggressive marketing strategies, further compounds the problem of child food poverty [23]. In Ethiopia, the consumption of energy-dense but nutrient-poor foods is increasing, displacing traditional diets that are often more nutritious. This shift is driven by globalization, and the influence of food marketing, which disproportionately targets low-income households [24]. In the Sidama Region, these trends may undermine efforts to improve child nutrition and worsen dietary inequalities [25].

The Sidama Region in southern Ethiopia is predominantly rural, with a high reliance on subsistence agriculture [26]. Despite its fertile land and agricultural potential, the region faces significant challenges, including high poverty rates, food insecurity, and limited access to healthcare services [18]. Child malnutrition remains a pressing issue, with high rates of stunting and wasting reported among children under five [27,28]. The region's limited infrastructure and resources, together with its susceptibility to climate change, make efforts to alleviate CFP prevalence even more difficult [29].

Although CFP prevalence is becoming more widely acknowledged as a serious public health concern, little is known about how common it is and what factors contribute to it in the Sidama Region, specifically in the Dale District, among children ages 6–23 months. Despite being particularly susceptible to the negative effects of malnutrition, this age group is still not well understood in the local environment. In order to assess the prevalence of CFP and the factors that lead to it, this study examined children aged 6–23 months who lived in the Dale district in the Sidama region of Ethiopia.

## Materials and methods

### Study area

The study was conducted in selected kebeles of the Dale district, which is in the Sidama Region in Southern Ethiopia. Dale district is bordered by Wonsho woreda in the East, Loka Abaya woredas in the West, Aleta Wondo and Aleta Chuko in the South, Darara district in the northwest, and Shebedino in the North. The district has a total of 32 rural and 4 urban kebeles which were the lowest administrative unit in Ethiopia. The main town in the district, Yirgalem town is located 320 km from Addis Ababa and 42 km to the South of Hawassa city.

The total population of the district was estimated at 272,203 with 50,113 households and it is among the most densely populated districts in the country. The district has 33 health posts, 10 health centers, and one general hospital [30]. The

majority of the district's rural kebeles have at least one health post run by health extension personnel. Farmers who grew ensete (Ensete ventricosum) known as a *false banana* for its resemblance to the domesticated banana plant, is a staple food for over 20 million people in the densely populated region of Ethiopia., maize, kale, cabbage, and haricot beans lived in the study area. In the region, economic crops like coffee, khat (Catha edulis), and fruits are produced. Sheep, goats, and cows are among the other animals raised in the district. The productive safety net program (PSNP), a non-governmental initiative aimed at addressing extreme household food insecurity, is only available in a small number of kebeles in the area. Among the following listed 36 kebeles six (Soyama, Dela Naramo, Shoye, Wara, Bera Chale, Debub Mesenkela) of them were selected randomly (Fig 1).

### Study design and period

A community-based cross-sectional study was employed. The study was conducted from February 15 to March 15, 2024.

### Source and study populations

All mothers or caretakers who had children aged 6–23 months lived in the Dale district Sidama Region was the source population. Randomly selected mothers or caretakers who had children aged 6–23 months from the selected kebeles of Dale district during the data collection period were the study population.

### Inclusion and exclusion criteria

All mothers or caretakers of children aged 6–23 months who lived in the Dale district for at least six months were included in the study. Mothers or caretakers of a child who were seriously ill and unable to communicate during the data collection period were excluded from the study.

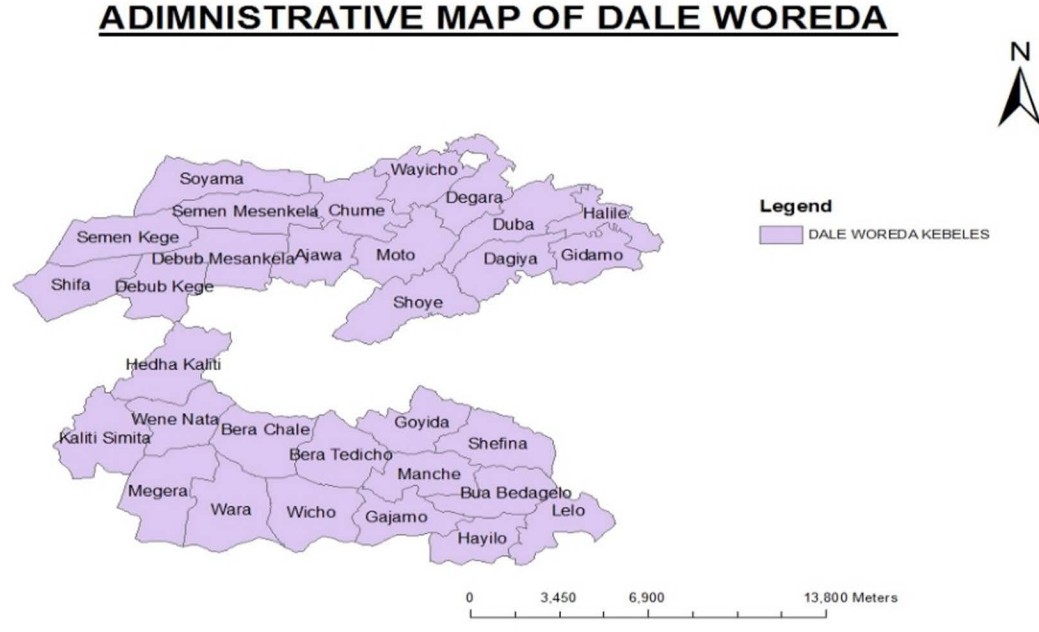

**Fig 1. Administrative map of Dale district, Sidama region, Ethiopia.**

## Sample size determination

**The sample size for the first objective.** The sample size for the first objective was calculated by using a single population proportion formula with the following assumptions:

- 95% level of confidence (Z = 1.96)

- p = 81.8% prevalence of inadequate dietary diversity from a study done in Northwest Ethiopia [31].

- The margin of error 5% (d = 0.05)

- α = the level of significance

- n $=$ $(Z1 - \alpha/2)^2$ p $(1 - p)/d^2$

- n $= \frac{(1.96)^2 * 0.818 * 0.182}{(0.05)^2} = 229$

The sample size determined for the first specific objective was 229.

**The sample size for the second objective.** The sample size was also calculated for the second specific objective by using Epi Info version 7.2 Statcalc based on the findings from studies conducted in Ethiopia, considering a 95% confidence level, power of 80%, and 1:1 ratio (Table 1).

Considering that the sample size calculated for the first specific objective was larger than the second objective, a sample size of 229. To account for the multistage sampling design, a design effect (deff = 2) was incorporated, resulting in a required sample size of 458. Additionally, with accounting for the 90% response rate the minimum final sample size was 509.

## Sampling procedures

A multi-stage random sampling technique was employed for this study. Initially, from the total of 36 kebeles in Dale district, six kebeles (the smallest administrative units in Ethiopia) were selected using a simple random sampling technique. Subsequently, the total sample size was allocated based on probability proportional to the size of households with children aged 6–23 months in each selected kebele. To establish the sampling frame, data were compiled from the community health information system and lactating women registers available at health posts within the randomly selected kebeles. From the prepared sampling frame, mothers having children aged 6–23 months selection was conducted through a simple random sampling technique. In instances where multiple eligible children reside within the same household during data collection, one child was randomly selected for inclusion in the study (Fig 2).

## Data collection instruments and procedures

Utilizing a structured interviewer-administered questionnaire that was modified from earlier research, data was gathered from study participants [36–39]. Five components make up the data collecting tool: sociodemographic parameters (father, mother, and child), factors connected to reproductive health and health services, factors related to the household, factors related to knowledge and attitude, and items for evaluating supplementary feeding practices. To ensure uniformity, the

**Table 1. Sample size determination for factors associated with inadequate dietary diversity.**

| Variable | CL | Power | % in unexposed | Ratio | AOR | n | Reference |
|---|---|---|---|---|---|---|---|
| Maternal education | 95% | 80% | 71.7 | 1:1 | 4.49 | 132 | [32] |
| Media exposure | 95% | 80% | 65.4 | 1:1 | 5.22 | 96 | [33] |
| Maternal age | 95% | 80% | 70 | 1:1 | 0.44 | 222 | [34] |
| Knowledge dietary diversity | 95% | 80% | 70.1 | 1:1 | 0.35 | 136 | [35] |

surveys were initially created in English, translated into Sidamu afo, and then returned to English. Data were gathered using the Sidamu afo language questionnaire. After that, the KoBo Toolbox was used to code the questionnaire for effective electronic data collection.

Poor child food practice was assessed using a structured tool that was adapted from WHO core indicators which was used to assess complementary feeding practices of infants and young children in similar settings [39]. The tool consists of three sections: introduction of solid, semi-solid, or soft foods at 6–8 months of age, meal frequency, and dietary diversity. Complementary feeding was categorized as appropriate when the child satisfies all three factors timely introduction, a minimum number of meals per day, and a minimal amount of dietary variety while it was deemed inappropriate if the child satisfies either of the indicators.

A standardized, structured, and validated Household Food Insecurity Access Scale (HFIAS) was used to assess the level of household food insecurity under household-related factors. The tool consisted of nine questions, based on the frequency of occurrence over the past 30 days, participants' scores ranged from 0 to 27. A higher HFIAS score indicates more inadequate access to food and greater household food insecurity, while a score of 0 indicates secure access to food [40]. Knowledge towards complementary feeding was assessed using 10 items, each correct response will be scored 1, and an incorrect response will be scored 0 [36]. The mean value was used to categorize respondents as having good knowledge or poor knowledge. Attitude towards complementary feeding was assessed through eight-item questions. Five-point Likert scale response, wherein a score of 5 = strongly agree, 4 = agree, 3 = neutral, 2 = disagree, and 1 = strongly disagree [41]. A mean score was used to categorize the mothers or caregivers as having positive and negative attitudes.

Data collection involved four data collectors from public health offices and one supervisor with experience overseeing data collecting. The supervisor and data collectors attended a two-day orientation that discussed the study's goal, advantages, individual rights, informed consent, interviewing methods, and how to use the Kobo application. The interview took place at the home of the mother or carer. The Kobo app on cell phones was utilized by the data collectors to gather information. The supervisor helped the data collectors each day and made sure every questionnaire was consistent. The primary investigator oversaw and directed the whole data collection procedure.

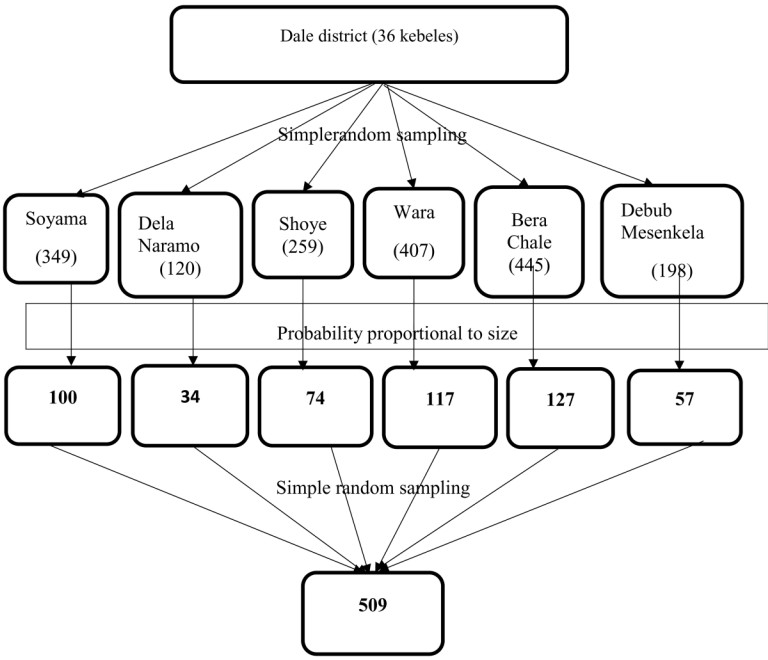

**Fig 2. Graphic presentation of the sampling procedure in Dale district, 2024.**

## Variables of study

### Dependent variable.

- CFP prevalence

### Independent variables.

- **Socio-demographic factors:** Age of child, sex of child, birth order, maternal age, marital status, maternal educational level, maternal occupation, father's age, father's educational level, father's occupation and residence.

- **Reproductive health and health service-related factors:** Number of live births, pregnancy status, ANC follow-up, number of ANC visits, place of delivery, mode of delivery, PNC follow-up, and pregnancy interval.

- **Household and community-related factors:** Family size, family type, number of children, food preference, household income, and household food security.

- **Knowledge and attitude-related factors:** Knowledge of complementary feeding, attitude towards complementary feeding, and source of information.

## Operational definitions

**Appropriate complementary feeding practice:** complementary feeding practice of children aged 6–23 months that fulfills both timely introduction and minimum dietary diversity and minimum meal frequency [36,37,39].

**Attitude:** The mother or caregiver was considered as having a positive attitude if she scored more than the mean score out of the nine attitude questions and if she scored below or equal to the mean, she will be considered as having a negative attitude [41].

**CFP prevalence** is measured using the UNICEF and WHO dietary diversity score. To meet the minimum dietary diversity for healthy growth and development, children need to consume food from at least five out of the eight defined food groups (1. breastmilk, 2. grains, roots, tubers, and plantains, 3. pulses, nuts, and seeds, 4. dairy products, 5. flesh foods (meat, poultry, and fish), 6. eggs, 7. vitamin A-rich fruits and vegetables, and 8. other fruits and vegetables). If children were fed: 0–2 food groups per day they are living in severe child food poverty, 3–4 food groups per day they are living in moderate child food poverty and 5 or more food groups per day they are not living in child food poverty [12].

**Food insecure:** a household that experienced 2–10 food insecurity indicators was considered as mildly food insecure household, 11–17 food insecurity indicators were considered as moderately food insecure household, and > 17 food insecurity indicators were considered as severely food insecure households [40].

**Food secure:** Households that experienced none of the food insecurity (access) conditions or just experienced worry, but rarely in the past 4 weeks were labeled as 'food secured or food secure households who experienced fewer than the first two food insecurity indicators [40].

**Inappropriate complementary feeding practice:** Complementary feeding practice of children aged 6–23 months that failed to fulfill either the timely introduction or minimum dietary diversity or minimum meal frequency [36,37,39].

**Kebele**: The lowest administrative unit in Ethiopia.

**Knowledge:** The mother or caregiver was considered as having good knowledge if she correctly answered above the mean score of knowledge assessing questions and poor knowledge if she answered equal and below the mean score of knowledge assessing questions [36].

**Minimum acceptable diet:** The proportion of children aged 6–23 months who were breastfed and who got at least two milk feedings, as well as the percentage of non-breastfed children aged 6–23 months who received at least two milk feedings and had at least the minimum dietary diversity (excluding milk feeds) and minimum meal frequency during the previous day [11,12].

**Minimum dietary diversity:** This study defined children aged 6–23 months who received foods from five or more food groups from eight food groups [11,12].

**Minimum meal frequency:** children receiving complementary foods (solid, semi-solid, or soft foods) a minimum number of times or more among children 6–23 months. The recommended number of meals per day for 6–8 months, 9–11 months & 12–23 months is 2–3 times, 3–4 times and 3–4 plus 1–2 snacks respectively [11,12].

**Timely introduction of complementary feeding:** In this study, children aged 6–23 months started complementary foods (solid, semi-solid, or soft foods) at the age of 6 months [11,39].

## Data quality control

One week before the actual data collecting time, a pretest was carried out with 5% of the sample size to identify and address potential limitations or shortcomings in the data collection instruments and data collection process, ensuring the quality of the data collected during the main study. Based on the results of the pretest, the questionnaire was reviewed for clarity, understandability, uniformity, and completeness. The review process is iterative, involving multiple rounds of revisions and refinements based on the feedback received from the pretest participants. The goal was to ensure that the final questionnaire was clear, understandable, uniform in interpretation, and comprehensive in capturing the necessary data for the study.

Additionally, before the commencement of data collection, both the data collectors and supervisors received two-day training. The training covered a range of important topics including study objectives, ethical considerations, data collection procedure, questionnaire administration, interview techniques, Kobo application utilization, and data quality control. Additionally, the supervisors were received training on how to evaluate the consistency and thoroughness of surveys. Additionally, the questionnaire was properly coded and categorized. By properly coding and categorizing the questionnaire responses, it streamlined data management, enhanced accuracy and consistency, facilitated data analysis, improved data interpretation, and maintained data quality throughout the research process.

## Data entry and analysis

The data collected in the Kobo Toolbox was exported to Statistical Software for Social Sciences (SPSS) version 27 for cleaning and analysis. Descriptive statistics (frequency, mean, standard deviation, and percentage) were used to describe the participants and determine the prevalence of inappropriate complementary feeding practices. Bivariable and multivariable analysis using logistic regression was used to identify the determinants of inappropriate complementary feeding practices. Variables having a value $P \leq 0.25$ during bivariable analysis were entered into multiple logistic regression models for control of confounding. The fitness of the logistic regression model was assessed by the Hosmer & Lemeshow test ($p = 0.894$) and variance inflaion factor (VIF < 10) was used to check multicollinearity between to independent variables. Adjusted Odds ratios (AOR) at 95% confidence intervals (CI) and P-values were used to present the findings. A P-value less than 0.05 was used as the criterion for statistical significance.

## Ethical consideration

Human subjects were used in the study, and ethical issues were considered. Before conducting the study, permission was requested from the Yirgalem Hospital Medical College School of Graduate Studies institutional ethical research board committee/IRB letter, which was acquired with reference number YHMC/IRB003 on November 2, 2023. Each of the selected kebele authority where the study was performed received a letter of cooperation, the Dale district administration received an official letter authorizing the project's commencement, and data collection began with their approval.

At the household level, the respondents' written and informed oral consent was requested and took. The volunteers were given a comprehensive description of the study's goals, methodology, duration, possible risks, and benefits in the

local language. In addition to being guaranteed anonymity, respondents were informed that their data would only be utilized for study.

By using unique code numbers instead of respondent names on the questionnaires, confidentiality was guaranteed for analysis.

## Results

### Socio-demographic characteristics

The study included 503 children, with a response rate of 98.8%. The mean (SD) age of the children was 12.8 (±5.3) months and 256(50.9%) were aged 12–23 months. There were 255(50.7%) female children. Almost all, 497 (98.8%) of the mothers were married. Two hundred forty-one (48.1%) of the mothers had no formal education and 429 (85.3%) of the mothers were housewives. Many families resided in rural areas 488(97%) (Table 2).

### Reproductive health and health service-related factors

The majority, 346 (68.8%) of the mothers had two or fewer children and planned their pregnancies (485, 96.4%). Antenatal care (ANC) was widespread at 481 (95.6%), with 225 (46.8%) having four or more visits. Most deliveries occurred at health centers (68.6%), and vaginal deliveries were predominant (93.0%). Postnatal care (PNC) follow-up was high (89.7%), and almost all mothers received complementary feeding counseling (99.6%). For those with multiple children, 61.0% had a birth interval greater than 23 months (Table 3).

### Household and community-related factors

The mean family size (SD) of the households was 4.3 (± 1.4) members. Most families had 5 or fewer members 408(81.1%) and one child under five 361(71.8%). Cultural food preferences for children were reported by 156(31.0%) of respondents. The majority lived with their husband and children 379(75.3%). Over half 281(55.9%) had a monthly income of less than 22.4$. Home gardening 498(98.8%) and crop production 442(87.9%) were very common. Many owned small livestock 451(89.7%), while fewer owned large livestock 225(44.7%). Regarding the household food security scale, 357 (69.8%) of the mothers were from food-insecure households while less than one-third 152(30.2%) were from food-secure households (Table 4).

### Knowledge and attitude-related factors

The participants' mean (SD) knowledge score was 7.4 (± 1.6), and 328 (65.2%) of them showed that they knew a lot about complementary feeding. Accordingly, 275 participants (54.7%) had unfavourable attitudes, whereas the mean (SD) attitude score was 39.1 (± 3.9). Health professionals accounted for 449 (55.0%) of the information sources, followed by family members 205 (25.2%) and friends 72 (8.8%) (Table 5).

### The magnitude of child food poverty (CFP) prevalence

The magnitude of CFP prevalence at 30%, inadequate acceptable diet (IAD) at 41%, and inadequate meal frequency (IMF) at 23%. The timely introduction of complementary feeding occurred in 76.9% (n = 387) of children. In the reverse the minimum meal frequency was achieved by 77% (n = 388) of children, while 70% (n = 353) met the minimum dietary diversity requirement (Figs 3 and 4).

### Factors associated with CFP prevalence

During the bivariable logistic regression analysis, several factors were significantly associated with CFP prevalence. These included the age of the child, age of the mother, educational status of the mother, planned pregnancy, postnatal care (PNC) follow-up, family size, income, household food security, knowledge, and attitude (Table 6).

**Table 2. Socio-demographic characteristics of households having children aged 6 to 23 months in Dale district, Sidama region, Ethiopia, 2024.**

| Variable | Category | Frequency | Percent |
|---|---|---|---|
| **Child age (months)** | 6–11 | 247 | 49.1 |
| | 12-23 | 256 | 50.9 |
| **Birth order of the child** | First | 210 | 41.7 |
| | Not first | 293 | 58.3 |
| **Mother's age (in completed years)** | ≤ 24 | 112 | 22.3 |
| | 25-29 | 251 | 49.9 |
| | 30-34 | 99 | 19.7 |
| | ≥ 35 | 41 | 8.2 |
| **Marital status** | Single | 6 | 1.2 |
| | Married | 497 | 98.8 |
| **Mother's educational status** | No formal education | 242 | 48.1 |
| | Primary education | 143 | 28.4 |
| | Secondary education | 90 | 17.9 |
| | College or above | 28 | 5.6 |
| **Mother's occupation** | Housewife | 429 | 85.3 |
| | Employee | 16 | 3.2 |
| | Daily laborer | 13 | 2.6 |
| | Merchant | 40 | 8.0 |
| | Other | 5 | 1.0 |
| **Father's age (years)** | ≤ 29 | 100 | 19.9 |
| | 30–34 | 128 | 25.4 |
| | 35-39 | 167 | 33.2 |
| | ≥ 40 | 108 | 21.5 |
| **Father's educational status** | No formal education | 144 | 28.6 |
| | Primary education | 151 | 30.0 |
| | Secondary education | 135 | 26.8 |
| | College or above | 73 | 14.5 |
| **Occupation of the father** | Farmer | 234 | 46.5 |
| | Employee | 60 | 11.9 |
| | Daily laborer | 31 | 6.2 |
| | Merchant | 170 | 33.8 |
| | Other | 8 | 1.6 |

After adjusting for potential confounding variables in multivariable logistic regression the age of the child, sex of the child, number of ANC follow-ups, family size, and knowledge remained as independent variables for CFP prevalence (Table 6).

The analysis revealed that children aged 6–11 months had 3.022 times higher odds of experiencing CFP prevalence compared to children aged 12–23 months (AOR = 3.022, 95% CI: 1.813, 5.037) (Table 6).

Mothers with female children had 1.783 times higher odds of engaging in CFP prevalence compared to those having male children (AOR = 1.783, 95% CI: 1.123, 2.830) (Table 6).

Mothers having more than five family sizes had 3.715 times higher odds of engaging in CFP prevalence compared to those having five and below (AOR = 3.715, 95% CI: 1.974–6.993) (Table 6).

**Table 3. Reproductive health and health service-related factors of the mothers having children aged 6 to 23 months in Dale district, Sidama region, Ethiopia, 2024.**

| Variable | Category | Frequency | Percent |
|---|---|---|---|
| Parity | ≤ 2 | 346 | 68.8 |
| | > 2 | 157 | 31.2 |
| Pregnancy planned | Yes | 485 | 96.4 |
| | No | 18 | 3.6 |
| ANC follow up | Yes | 481 | 95.6 |
| | No | 22 | 4.4 |
| Number of ANC visits | One visit | 39 | 8.1 |
| | Two visits | 34 | 7.1 |
| | Three visits | 183 | 38.0 |
| | Four or more | 225 | 46.8 |
| Place of delivery | Hospital | 120 | 23.8 |
| | Health Center | 347 | 69.0 |
| | Home | 36 | 7.2 |
| Mode of delivery | Vaginal | 468 | 93.0 |
| | CS | 35 | 7.0 |
| PNC follow up | Yes | 451 | 89.7 |
| | No | 52 | 10.3 |
| CF counselling | Yes | 449 | 99.6 |
| | No | 2 | 0.4 |
| Birth interval (months) (N = 292) | ≤ 23 | 114 | 39.0 |
| | > 23 | 178 | 61.0 |

Additionally, the odds of CFP prevalence were 2.436 times higher among mothers who did not have above four times ANC follow-up compared to those who did (AOR = 2.436, 95% CI: 1.429, 4.152) (Table 6).

Furthermore, mothers with inadequate knowledge had 6.23 times higher odds of CFP prevalence compared to mothers with adequate knowledge (AOR = 5.148, 95% CI: 3.146–8.426) (Table 6).

## Discussion

This study found that the prevalence of CFP among children aged 6–23 months in Dale District, Sidama Region, Ethiopia was 29.8% (95% CI: 26–34). The prevalence of timely introduction of complementary feeding, MMF, and MDD, the MAD was 76.9%, 77.1%, 70.2%, and 59% respectively. Child age, child sex, number of antenatal care follow-up, family size within the household, and maternal knowledge were among the factors that the study found to be substantially linked with the prevalence of CFP.

This study found that 29.8% of children aged 6–23 months in the Dale District, Sidama Region, Ethiopia, were living in food poverty. This result was less than that of research done in other regions of Ethiopia, such the Dejen district (86.4%) [32], Dabat district (83%) [42], Haramaya district (80.3%) [43], Addis Ababa (40.1%) [32], Awi zone (46.6%) [44], Mareka district (61%) [45], Goriche district (89.4%) [46], and Aleta Wondo district (88%) [47] in Sidama region, and Ethiopia from 2016 EDHS (85.1%) [20]. These variations could be attributed to several factors, including methodological differences in defining and measuring CFP prevalence, geographical and socio-economic variations across study areas, temporal factors reflecting changes in nutritional practices over time, and differences in sample characteristics and data collection methods. The lower prevalence in this study compared to some others might indicate improvements in dietary diversity in the Dale District, possibly due to successful health education and nutrition counseling programs, better access to diverse foods, or increased awareness about preventing CFP prevalence.

**Table 4. Factors pertaining to homes and communities with children aged 6 to 23 months in Dale district, Sidama region, Ethiopia, 2024.**

| Variable | Category | Frequency | Percent |
|---|---|---|---|
| Family size | ≤ 5 | 408 | 81.1 |
| | > 5 | 95 | 18.9 |
| Number of under-five children | 1 | 361 | 71.8 |
| | 2 | 134 | 26.6 |
| | 3 | 8 | 1.6 |
| Cultural food preference for children | Yes | 156 | 31.0 |
| | No | 347 | 69.0 |
| Living condition | Husband and children | 379 | 75.3 |
| | Parents of mother/father | 94 | 18.7 |
| | Other relatives | 23 | 4.6 |
| | Other | 7 | 1.4 |
| Income in $ | <22.4 | 281 | 55.9 |
| | 22.4–44.8 | 187 | 37.2 |
| | > 44.8 | 35 | 7.0 |
| Practice home gardening | Yes | 497 | 98.8 |
| | No | 6 | 1.2 |
| Crop production | Yes | 442 | 87.9 |
| | No | 61 | 12.1 |
| Own small livestock (goats, sheep, hen) | Yes | 451 | 89.7 |
| | No | 52 | 10.3 |
| Own large livestock (cows, oxen) | Yes | 225 | 44.7 |
| | No | 278 | 55.3 |
| Household food security status | Food secure | 152 | 30.2 |
| | Food insecure | 357 | 69.8 |

**Table 5. Knowledge and attitude-related factors among mothers having children aged 6 to 23 months in Dale district, Sidama region, Ethiopia, 2024.**

| Variable | Category | Frequency | Percent |
|---|---|---|---|
| Knowledge | Inadequate | 175 | 34.8 |
| | Adequate | 328 | 65.2 |
| Attitude | Unvaourable | 275 | 54.7 |
| | Favorable | 228 | 45.3 |
| Source of information | Health professionals | 449 | 55.0 |
| | Family member | 205 | 25.2 |
| | Friends | 72 | 8.8 |
| | Mass media | 48 | 5.9 |
| | Community health worker | 42 | 5.1 |

In this study, the prevalence of timely introduction of complementary feeding was 76.9%, MMF was 77.1%, MDD was 70.2%, and MAD (59%) higher than figures reported in previous studies in Ethiopia and other low- and middle-income countries. For example, a study in Damot Weydie District, Southern Ethiopia, reported only 50.6% of mothers initiated complementary feeding at the recommended 6 months [36], and UNICEF's 2022 data showed only 50% met the MMF, and 31% achieved MDD [48]. These higher rates in our study may be due to better access to antenatal care follow-ups,

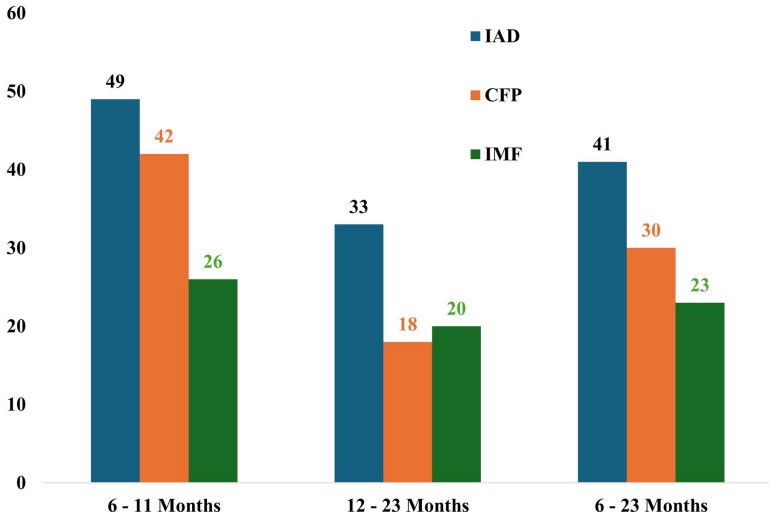

and successful nutrition interventions. While the high rates of timely introduction and MFF are promising, MDD and MAD still require improvement, likely influenced by socioeconomic status, and maternal knowledge.

This study found that children aged 6–11 months had about three times higher odds of experiencing CFP prevalence compared to children aged 12–23 months. This finding is consistent with studies conducted in Shashemene town, southern Ethiopia [49], Arsi Negele [50], Tanzania [51], Pakistan [52], and West China [53]. This association can be attributed to several factors: younger infants are in a transitional period from exclusive breastfeeding to optimal complementary feeding, and mothers may face challenges in introducing new foods while maintaining breastfeeding. Additionally, caregivers might be less confident in preparing and offering a variety of foods to younger infants, leading to less diverse diets. As children grow older, caregivers generally become more experienced and knowledgeable about feeding practices, which could explain the improved practices in the older age group.

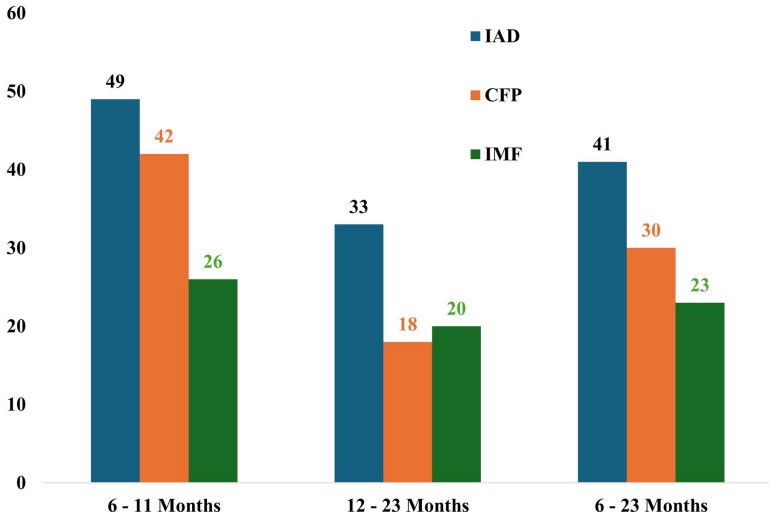

**Fig 3. CFP prevalence, IMF, and IAD by age in Dale District 2024.**

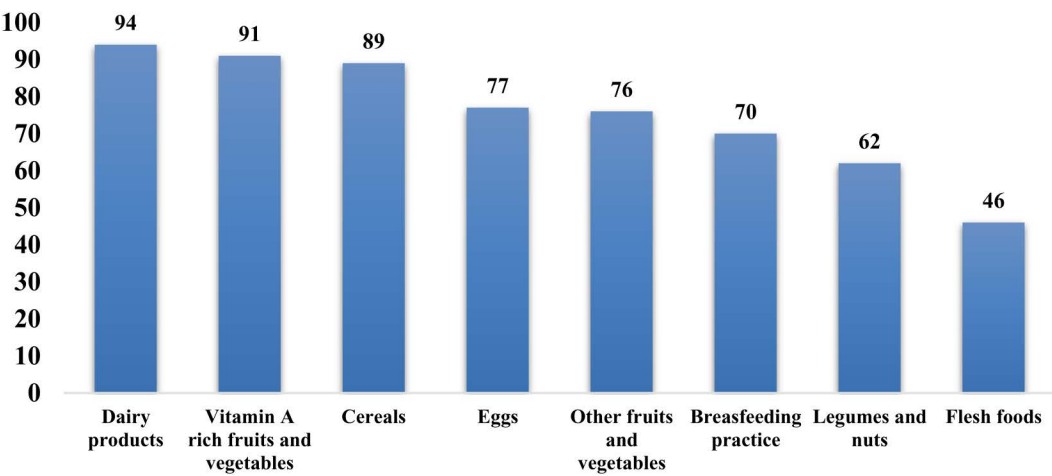

**Fig 4. Food group intake percentage to children using 24 recalls in Dale District 2024.**

The odds of CFP prevalence were approximately four times higher among women who did not have antenatal care (ANC) follow-up compared to those who did. Compared to women who did not get infant and young child feeding (IYCF) counseling and ANC visits, the food dietary diversity of their children was better for those mothers. These results have been corroborated by further regional investigations [54–57]. ANC visits provide the right counseling to help parents improve their behaviors regarding their children's dietary diversity. As a result, the study found that ANC data is positively significant for CFP prevalence. This data backs up our conclusions [54,58]. Therefore, health education and nutrition counseling for mothers is an important activity in our nation; it should be done regularly to improve children's food dietary diversity. ANC visits provide crucial opportunities for healthcare providers to educate mothers about optimal IYCF practices. During these visits, mothers receive guidance on when and how to introduce optimal complementary foods, the

**Table 6. Factors associated with CFP prevalence among children aged 6 to 23 months in Dale district, Sidama region, Ethiopia, 2024.**

| Variables | CFP prevalence | | COR (95% CI) | AOR (95% CI) | P – value |
|---|---|---|---|---|---|
| | Yes | No | | | |
| **Child age** | | | | | |
| 6–11 | 104 | 143 | 3.12(2.21- 4.99) | **3.022(1.813-5.037)\*\*** | **0.000** |
| 12–23 | 46 | 210 | 1 | 1 | |
| **Sex of the child** | | | | | |
| Male | 63 | 185 | 1 | 1 | |
| Female | 87 | 168 | 1.521(1.034-2.236) | **1.783(1.123-2.830)\*** | **0.014** |
| **Maternal education** | | | | | |
| No formal education | 91 | 151 | 2.784(1.624-4.771) | 1.279(0.644-2.542) | 0.482 |
| Primary (1–8) | 38 | 105 | 1.672(0.917-3.046) | 1.329(0.633-2.790) | 0.452 |
| Secondary and above | 21 | 97 | 1 | 1 | |
| **Number of antenatal care follow-up** | | | | | |
| Four and above four | 193 | 34 | 1 | 1 | |
| Below four | 160 | 116 | 3.772(2.426-5.866) | **2.436(1.429-4.152) \*** | **0.001** |
| **Place of delivery** | | | | | |
| Home | 21 | 15 | 3.668(1.834-7.335) | 1.805(0.747-4.363) | 0.189 |
| Health facility | 129 | 338 | 1 | 1 | |
| **Postnatal care follow up** | | | | | |
| Yes | 128 | 323 | 1 | 1 | |
| No | 22 | 30 | 1.851(1.029-3.328) | 1.530(0.670-3.496) | 0.313 |
| **Family size** | | | | | |
| ≤ 5 | 103 | 305 | 1 | 1 | |
| > 5 | 47 | 48 | 2.899(1.830-4.593) | **3.715(1.974-6.993)** | **0.000** |
| **Household food security status** | | | | | |
| Food secured | 30 | 122 | 1 | 1 | |
| Food insecure | 120 | 231 | 2.113(1.339-3.334) | 1.681(0.922-3.065) | 0.09 |
| **Knowledge** | | | | | |
| Inadequate | 98 | 77 | 6.755(4.436-10.287) | **5.148(3.146-8.426)\*\*** | **0.000** |
| Adequate | 52 | 276 | 1 | 1 | |
| **Attitude** | | | | | |
| Unfavorable | 92 | 183 | 1.474(0.998-2.175) | 1.498(0.907-2.474) | 0.115 |
| Favorable | 58 | 170 | 1 | 1 | |

*p-value<0.05, **p-value<0.0001.

importance of dietary diversity, and appropriate MMF. Moreover, ANC follow-ups allow healthcare providers to address any feeding challenges the mother might be experiencing the child's growth and development [59].

According to this study, the likelihood of CFP prevalence was approximately four times higher for families with more than five members than for children between the ages of 12 and 23 months. This result is in line with research done in Gurage zone [60] and Konso zone [61] southern Ethiopia Pakistan [52]. This could be the result of growing family numbers, which raises the demand for the recommended diet and, eventually, could result in a household-level diet deficit. It further clarified that one important mitigating method is to distribute food among household members equitably based on their physiological demands.

This study showed that the prevalence of CFP is correlated with being a female. Girls are less socially and nutritionally preferred in some cultures, which may be due to male favoritism. This study, on the other hand, contradicted research done in Ethiopia, including Mekelle City [9], Tehuledere district [62], Jimma [63], Chiro town [64], and Wukro town [65] which confirmed that males were more affected due to thinness than girls. This variable includes several African countries including Pakitan [66].

Mothers with inadequate knowledge had five times the odds of having children living in food poverty compared to mothers with adequate knowledge. This conclusion is supported by an institutional cross-sectional study conducted in Addis Ababa, which showed that mothers who understood dietary diversity well were less likely to have children living in food poverty than mothers who did not [32]. Knowledge plays a crucial role in shaping feeding practices. Compared to their peers, mothers who were well-versed in parenting techniques and nutritional diversity were more likely to serve their children a variety of meals. According to a study conducted in southern Ethiopia, a unit increase in maternal awareness of IYCF was linked to increases in dietary variety scores of 0.41 [46] and 1.98 [32]. Additionally, mothers who received the IYCF message during ANC took part in a cookery demonstration [46] and were exposed to IYCF material in the media were more likely to consume a variety of foods [46,67]. According to a study conducted in Nairobi, Kenya [68], mothers' awareness of good eating behaviors is linked to a variety of child-feeding techniques. For proper child feeding practices, it is crucial to have a better awareness of the various food products and their advantages for a child's development and health. The health extension workers might do this. In addition to offering some curative treatments, their primary responsibility is to educate the community about the eighteen health extension packages (nutrition is one of them) [69] through home visits and at the health facility.

## Limitations of the study

Despite addressing CFP prevalence, this study has several limitations. CFP prevalence was age-specific with narrow age ranges, and typically assessed through caregiver reports, which could introduce recall bias and underestimate. In contrast to other observational research, cross-sectional studies don't keep track of participants across time. Furthermore, a causal association between CFP and the predictors cannot be established by cross-sectional study designs, such as the one employed in this investigation. Notwithstanding these drawbacks, the study's large random sample, intriguing results, and identification of areas requiring further research are all strengths.

## Conclusion

Child food poverty remains a major public health concern in Dale District, Sidama Region, Ethiopia, with a prevalence of 29.8% among children aged 6–23 months. Despite some positive gains, there is still much need for improvement in the following areas: MMF (77.1%), MDD (70.2%), MAD (59%) and the prevalence of timely introduction of complementary feeding (76.9%). The study found a number of important characteristics linked to the prevalence of CFP including younger child age (6–11 months), child sex, antenatal care follow-up, family size, and inadequate maternal knowledge to reduce CFP prevalence. These findings underscore the complex interplay of socio-demographic, health service utilization, and knowledge-related factors that brought CFP prevalence in the study area. Develop and implement comprehensive

nutrition education programs targeting mothers and or caregivers having children under two years to improve their knowledge to prevent CFP prevalence.

## Supporting information

**S1 Data. Child food poverty SPSS data.**
(SAV)

## Acknowledgments

The authors thank the School of Public Health, Yirgalem Hospital Medical College, supervisors, data collectors, and study participants.

## Author contributions

**Conceptualization:** Fentaw Wassie Feleke, Zenamlak Yemane, Assefa Philipos Kare, Amelo Bolka.

**Data curation:** Fentaw Wassie Feleke, Zenamlak Yemane, Assefa Philipos Kare, Amelo Bolka.

**Formal analysis:** Fentaw Wassie Feleke, Zenamlak Yemane, Assefa Philipos Kare, Amelo Bolka.

**Funding acquisition:** Fentaw Wassie Feleke, Zenamlak Yemane, Assefa Philipos Kare, Amelo Bolka.

**Investigation:** Fentaw Wassie Feleke, Zenamlak Yemane, Assefa Philipos Kare, Amelo Bolka.

**Methodology:** Fentaw Wassie Feleke, Zenamlak Yemane, Assefa Philipos Kare, Amelo Bolka.

**Project administration:** Zenamlak Yemane, Assefa Philipos Kare, Amelo Bolka.

**Resources:** Fentaw Wassie Feleke, Zenamlak Yemane, Assefa Philipos Kare, Amelo Bolka.

**Software:** Fentaw Wassie Feleke, Zenamlak Yemane, Assefa Philipos Kare, Amelo Bolka.

**Supervision:** Zenamlak Yemane.

**Validation:** Fentaw Wassie Feleke, Zenamlak Yemane, Assefa Philipos Kare, Amelo Bolka.

**Visualization:** Fentaw Wassie Feleke, Zenamlak Yemane, Assefa Philipos Kare, Amelo Bolka.

**Writing – original draft:** Fentaw Wassie Feleke, Amelo Bolka.

**Writing – review & editing:** Fentaw Wassie Feleke, Zenamlak Yemane, Assefa Philipos Kare, Amelo Bolka.

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
