## [Decision Letter · Decision Letter 0]

Dear Dr. Feleke,

Thank you for submitting your manuscript to PLOS ONE. After careful consideration, we feel that it has merit but does not fully meet PLOS ONE’s publication criteria as it currently stands. Therefore, we invite you to submit a revised version of the manuscript that addresses the points raised during the review process.

We look forward to receiving your revised manuscript.

Kind regards,

Dinaol Abdissa Fufa, Mph

Academic Editor

PLOS ONE

2. You have mentioned N/A in the ethics statement of the submission system while you have obtained ethics approval from the IRB. Kindly update the ethics approval information in the ethics statement of the submission system.

4. Please ensure that you refer to Figure 1 and 2, in your text as, if accepted, production will need this reference to link the reader to the figure.

5. Please remove all personal information, ensure that the data shared are in accordance with participant consent, and re-upload a fully anonymized data set.

6. We note you have included a table to which you do not refer in the text of your manuscript. Please ensure that you refer to Table 5 and 6, in your text; if accepted, production will need this reference to link the reader to the Table.

Reviewers' comments:

Reviewer's Responses to Questions

**Comments to the Author**

1. Is the manuscript technically sound, and do the data support the conclusions?

Reviewer #1: Yes

Reviewer #2: Yes

Reviewer #3: Partly

2. Has the statistical analysis been performed appropriately and rigorously?

Reviewer #1: Yes

Reviewer #2: Yes

Reviewer #3: No

3. Have the authors made all data underlying the findings in their manuscript fully available?

Reviewer #1: No

Reviewer #2: No

Reviewer #3: Yes

4. Is the manuscript presented in an intelligible fashion and written in standard English?

Reviewer #1: No

Reviewer #2: Yes

Reviewer #3: No

Reviewer #1: The authors did an impressive work in outlining and explaining the full details regarding their work. I have few points for authors to reconsider and edit:

1. In the first paragraph of the introduction, you stated "The main causes of this illness include...."

Child food poverty is NOT an illness but it can lead to sever illnesses

2. In the Second paragraph of the introduction, you mentioned the meaning of UNICEF acronym. however, this is not the full correct meaning

3. In Materials and methods, Study area, you mentioned kebeles. This word is specific to Ethiopia only and not well known around the world. therefore, it is preferable to mention what is the meaning of this word first and then you can use it freely.

4. In Materials and methods, Study area, second paragraph, the author mentioned Ensete ventricosum and Catha edulis. Please make sure they are written in italic (scientific names of plants)

5. Figure 1 should show the highlighted kebeles where the investigation took place

6. In Sample size determination, Sample size for first objective, state the meaning of α and p

7. In Variables of study, authors mentioned ANC and PNC for the first time here without explaining the meaning of the Acronyms. Please state their meaning

8. In Food insecure, HH was used to refer to household (maybe!!?). Please make it clear

9. In Data quality control, authors stated "Additionally, the supervisors will receive training on how to evaluate the consistency and thoroughness of surveys"

will receive is not appropriate here

10. In Data quality control, VIF was used without explaining its meaning

11. In table 4, please state the meaning of ETB here

12. In table 5, authors regarded parents AND family member as separate source of information. Why parents and family member are not in the same category? Do all pregnant women in this study have Family members besides parents?

13. In figure 3, authors used acronyms without prior explanation of their meaning in the previous paragraph

14. In the second page of the discussion, authors mentioned IYCF, MDD without prior explanation of their meaning in the previous paragraphs

15. In the second page of the discussion, authors should avoid using " in our nation"

16. In the third page of the discussion, authors need to clarify their statement.

This study demonstrated that being a female is associated with child food poverty. THIS IS THE FIRST LINE

This could be attributable to male patriotism (FAUVORISISM) instead of patriotism

in some communities where girls are less socially and nutritionally favored. This study was in line with studies conducted in Ethiopia such as Mekelle City [9], Tehuledere district(54), Jimma (55), Chiro town(56), and Wukro town (57) which confirmed that males were more affected due to thinness than girls. This variable includes several African countries including Pakitan (58). THE END OF THE PARAGRAPH DOES NOT AGREE WITH THE FIRST LINE

THIS PARAGRAPH CONTRADICT ITSELF because it mentioned that male ate more affected than female even though it stated different thing in the first line.

17. Authors mentioned a section called Abbreviation and or acronyms. Preferably, authors should state the meaning of their Abbreviation and or acronyms on their first appearance or mention in the manuscript to make the flow of the paper more easy to follow for the reader

18. in the reference section, authors referred to websites. In this case, authors should include the website url, and date of obtaining the data

Author should improve their manuscript by using English-editing service to avoid grammatical errors, etc....

Reviewer #2: This is a commendable and timely study addressing child food poverty in a high-need region of Ethiopia. This study provides important insights into a critical area of public health nutrition.

The identification of significant predictors—such as child age, sex, ANC follow-up, family size, and maternal knowledge—provides actionable insights for programmatic interventions. The discussion is well-supported with relevant literature and highlights practical implications.

Regarding the technicality of the manuscript, the manuscript is technically sound and presents a well-executed community-based cross-sectional study involving 503 participants. The research rigorously followed scientific procedures, and the results logically support the conclusions.

The statistical analysis is thorough and appropriate for a cross-sectional study. Also, the use of standardized indicators, validated tools, and rigorous statistical analysis adds credibility to the findings. These methods are robust and validate the associations claimed in the study.

The manuscript is well-written and easy to follow. The abstract, introduction, methods, results, and discussion are all structured logically and adhere to standard academic English.

Areas for Minor Improvement:

1) Consider clearly distinguishing between "child food poverty" and other related terms like "minimum dietary diversity" or "minimum acceptable diet."

2) Although ethical approval is mentioned, the Ethics Statement in the manuscript should be expanded. Explicitly mention:

a) Consent procedures (written/oral)

b) Ethical review body (Yirgalem Hospital Medical College)

c) Confidentiality measures (specify whether data were anonymized before analysis).

3) On Data Availability and Consent Statements, the manuscript says, “Data are available from the corresponding author upon request”, which contradicts PLOS ONE’s open data policy. Instead, the data should be made publicly available or archived in a repository, with links included.

4) The introduction provides a solid background, but the review can be improved by integrating more recent global data on child food poverty and nutrition transitions

5) While the introduction touches on the knowledge gap, the authors should emphasize more clearly what this study adds that others have not done in Sidama or Ethiopia more broadly.

6) Recommendations should be more action oriented. Rather than general calls to “raise awareness,” suggest specific programs or stakeholder groups.

7) There are occasional typos that should be corrected during revision, and they include:

a) foodv poverty in the conclusion rather than food poverty.

b) Abbreviations not abbrivation

c) ANC: Antenatal Care not Ante natal Care

d) AOR: Adjusted Odds Ratio not Adjust Odd Ratio

e) EBF: Exclusive Breastfeeding not E xclusive Breastfeeding (excessive spacing)

f) EDHS: Ethiopian Demographic and Health Survey not Ethiopian Demography and Health Survey

g) IDD: Inadequate Dietary Diversity not Iinadequte Dietary Diversity (double “I”)

h) MDGs: Millennium Development Goals not Millennium Development Goals’ (wrong apostrophe)

i) PNC: Postnatal Care not Post Natal Care

j) SPSS: Statistical Package for the Social Sciences not Statistical Package for the Social Science

8) I would recommend that acronyms be presented in alphabetical order for better readability.

9) A consistent colon and space format should be used especially in the abbreviation and/or acronym section as well as throughout the body of the article.

10) Use "to" instead of a dash for formal writing (February 15 to March 15, 2024) at the Study design and period.

Reviewer #3: Dear authors,

Please respond to the following:

1. English language editing is needed in some parts of the study.

2. The title is informative but too long and wordy. I suggest the following title: [Prevalence and Determinants of Child Food Poverty in Dale District, Sidama Region, Ethiopia].

3. Terminologies like “child food poverty” and “inadequate feeding” are used interchangeably, which may confuse readers. Define each key term clearly and consistently stick to one throughout the paper, especially in the abstract and introduction.

4. In Introduction: The study aim is broad and not broken down into measurable components. Try using the term (prevalence).

5. In Methods: The study period (February 15 – March 15, 2024) is listed, but seasonal variation that affects food availability is not discussed. Mention why this time frame was chosen and how it might affect feeding practices due to seasonal availability of food.

6. In Methods: The map and sampling diagram (Figure 1 and Figure 2) are mentioned, but not actually included in the provided text, and there's no description of what they show. Ensure all figures are labeled, included, and properly referenced in-text with brief captions.

7. In Methods: The sampling method may not be well explained. How were households selected? Were clusters used? Any stratification? Please respond clearly.

8. In limitations of the study: Limitations related to cross-sectional design, and generalizability are not addressed. Mention that causality cannot be inferred due to the study design.

**Do you want your identity to be public for this peer review?** For information about this choice, including consent withdrawal, please see our Privacy Policy

Reviewer #1: **Yes: ** Doaa Husham Majeed Alsaadi

Reviewer #2: **Yes: ** Chibuzor Stella Amadi

Reviewer #3: **Yes: ** Hayder Al-Momen

---

## [Author Response · Author response to Decision Letter 1]

1 May 2025

We appreciate the reviewers' insightful comments and suggestions on how to improve this work. As shown by the red colors, we, the authors, did our best to comprehend and consider each recommendation and observation made in the original work. Additionally, the work has been proofread for grammar and English usage. The web platform receives the final, corrected manuscript. We consider your wise, useful observation and suggestion. We have uploaded each reviewers comment response together with the the third version revised manuscript.

---

## [Decision Letter · Decision Letter 1]

Child food poverty prevalence and its associated factors in Dale district, Sidama region, Ethiopia

PONE-D-25-13161R1

Dear Dr. Feleke,

We’re pleased to inform you that your manuscript has been judged scientifically suitable for publication and will be formally accepted for publication once it meets all outstanding technical requirements.

Kind regards,

Dinaol Abdissa Fufa, Mph

Academic Editor

PLOS ONE

Additional Editor Comments (optional):

Reviewers' comments:

Reviewer's Responses to Questions

**Comments to the Author**

Reviewer #1: (No Response)

Reviewer #2: All comments have been addressed

Reviewer #3: (No Response)

2. Is the manuscript technically sound, and do the data support the conclusions?

Reviewer #1: Yes

Reviewer #2: Yes

Reviewer #3: Yes

3. Has the statistical analysis been performed appropriately and rigorously?

Reviewer #1: Yes

Reviewer #2: Yes

Reviewer #3: Yes

4. Have the authors made all data underlying the findings in their manuscript fully available?

Reviewer #1: Yes

Reviewer #2: Yes

Reviewer #3: Yes

5. Is the manuscript presented in an intelligible fashion and written in standard English?

Reviewer #1: No

Reviewer #2: Yes

Reviewer #3: Yes

Reviewer #1: The authors did a great job in addressing most of the comments of the reviewers which made the manuscript easy to follow and more understandable. the concept of the this study is interesting and valuable for future studies.

I have few points that i advice the authors to edit

1. Regarding plant scientific name, please make it italic for both Ensete ventricosum and Catha edulis.

2. For Figure 1, please highlight the selected six kebeles in different color (for example yellow) to make it easy to pinpoint their location and the distance between them in the map

other than these two suggestions, i think the manuscript is ready for publication

Reviewer #2: The authors have done an excellent job addressing the concerns and suggestions raised in the previous round of review. The revised manuscript demonstrates thoughtful revisions, particularly in areas related to ethical considerations and the correction of previously noted typographical errors.

This study makes a valuable contribution to the field of public health nutrition, offering meaningful insights through a well-executed community-based cross-sectional design. The research is both timely and relevant, addressing a public health concern with practical implications for policy and program planning.

The discussion section is well-developed, drawing effectively on relevant literature to support the interpretation of findings. It highlights the real-world applicability of the results, which strengthens the paper's contribution to public health practice.

From a methodological standpoint, the study remains technically sound. The statistical analyses are comprehensive, clearly described, and appropriate for the cross-sectional nature of the data. The authors have adequately controlled for potential confounding variables, enhancing the validity of their conclusions.

The manuscript is well-written, with a clear and logical structure that adheres to academic writing standards. Each section—the abstract, introduction, methodology, results, and discussion—is coherent, concise, and easy to follow, facilitating reader comprehension.

In summary, the authors have successfully strengthened the manuscript in response to prior feedback, and it is now suitable for publication. The study presents robust evidence and thoughtful interpretation, making it a meaningful addition to the literature in public health nutrition.

Reviewer #3: Dear authors,

Thank you for your responses.

I have expected you to address the suggestion of title change in a professional way.

Please consider the following title: [Prevalence and Determinants of Child Food Poverty in Dale District, Sidama Region, Ethiopia].

**Do you want your identity to be public for this peer review?** For information about this choice, including consent withdrawal, please see our Privacy Policy

Reviewer #1: **Yes: ** Doaa Husham Majeed Al-saadi

Reviewer #2: **Yes: ** Chibuzor Stella Amadi

Reviewer #3: No

---

## [Editor Report · Acceptance letter]

PONE-D-25-13161R1

PLOS ONE

Dear Dr. Feleke,

I'm pleased to inform you that your manuscript has been deemed suitable for publication in PLOS ONE. Congratulations! Your manuscript is now being handed over to our production team.

Kind regards,

on behalf of

Dr. Dinaol Abdissa Fufa

Academic Editor

PLOS ONE